# In Vivo Effects of Bay 11-7082 on Fibroid Growth and Gene Expression: A Preclinical Study

**DOI:** 10.3390/cells13131091

**Published:** 2024-06-24

**Authors:** Tsai-Der Chuang, Nhu Ton, Shawn Rysling, Omid Khorram

**Affiliations:** 1The Lundquist Institute for Biomedical Innovation, Torrance, CA 90502, USA; tchuang@lundquist.org (T.-D.C.); nhu.ton@lundquist.org (N.T.); shawn.rysling@lundquist.org (S.R.); 2Department of Obstetrics and Gynecology, David Geffen School of Medicine at University of California, Los Angeles, CA 90024, USA

**Keywords:** fibroid, cell proliferation, ECM accumulation, inflammation, xenografts

## Abstract

Current medical therapies for fibroids have major limitations due to their hypoestrogenic side effects. Based on our previous work showing the activation of NF-kB in fibroids, we hypothesized that inhibiting NF-kB in vivo would result in the shrinkage of tumors and reduced inflammation. Fibroid xenografts were implanted in SCID mice and treated daily with Bay 11-7082 (Bay) or vehicle for two months. Bay treatment led to a 50% reduction in tumor weight. RNAseq revealed decreased expression of genes related to cell proliferation, inflammation, extracellular matrix (ECM) composition, and growth factor expression. Validation through qRT-PCR, Western blotting, ELISA, and immunohistochemistry (IHC) confirmed these findings. Bay treatment reduced mRNA expression of cell cycle regulators (*CCND1*, *E2F1*, and *CKS2*), inflammatory markers (*SPARC*, *TDO2*, *MYD88*, *TLR3*, *TLR6*, *IL6*, *TNFα*, *TNFRSF11A*, and *IL1β*), ECM remodelers (*COL3A1*, *FN1*, *LOX*, and *TGFβ3*), growth factors (*PRL*, *PDGFA*, and *VEGFC*), progesterone receptor, and miR-29c and miR-200c. Collagen levels were reduced in Bay-treated xenografts. Western blotting and IHC showed decreased protein abundance in certain ECM components and inflammatory markers, but not cleaved caspase three. Ki67, CCND1, and E2F1 expression decreased with Bay treatment. This preclinical study suggests NF-kB inhibition as an effective fibroid treatment, suppressing genes involved in proliferation, inflammation, and ECM remodeling.

## 1. Introduction

Fibroids are benign tumors that affect a significant number of women, causing pain, abnormal uterine bleeding, and infertility [1]. The pathogenesis of these tumors, whose growth is regulated by ovarian steroids, has been under intense investigation, with inflammation playing a pivotal role in this process [2]. Several studies reported a marked dysregulation of profibrotic and proinflammatory cytokines and chemokines in the tumors or circulation of patients with fibroids [3,4]. The onset of excess inflammation may be an early event in tumorigenesis, as progenitor fibroid cells were demonstrated to secrete higher levels of proinflammatory cytokines [5]. A source of these inflammatory cytokines is the macrophages [6]. 

Work from our laboratory has focused on the role of NF-kB, the master regulator of inflammatory responses in fibroid pathogenesis. Aberrant NF-kB signaling has been implicated in a multitude of disorders, including inflammatory and autoimmune disorders and cancer [7]. The NF-kB complex is composed of a family of inducible transcription factors, including NF-kB1 (p50), NF-kB2 (p52), RelA (p65), RelB, and c-Rel. These transcription factors bind to a specific DNA site called kB enhancer as homo- or heterodimers, inducing transcription of proinflammatory genes and are normally sequestered in the cytoplasm by a family of inhibitory IKB proteins [8]. The activation of NF-kB involves both the canonical and noncanonical pathways. The canonical pathway activation results in quick and transient transcription activity, whereas the activation of the noncanonical pathway is slower and occurs through several TNF receptor superfamily members [8,9]. In earlier works, we reported the activation of NF-kB in fibroids, as evidenced by elevation of phosphorylated p65. This activation of NF-kB led to the aberrant expression of miR-29c, which regulates a wide range of extracellular matrix (ECM) genes that are highly overexpressed in fibroids [10]. We also reported increased phosphorylation of IKBKB in fibroids, which induces phosphorylation of IkBα and its proteasome degradation. This process allows NF-kB dissociation and its nuclear translocation. Additionally, we showed that miR-200c, which is also aberrantly expressed in fibroids, regulated IL8 by targeting IKBKB [11]. Based on our findings and those of others indicating that fibroid tumors are characterized by increased inflammation [12] and activation of NF-kB, we hypothesized that Bay 11-7082 (Bay), an inhibitor of inflammation, could be beneficial for the treatment of fibroids by reducing the expression of proinflammatory genes and inhibiting their growth. To test this hypothesis, we administered Bay over a two-month course in a human fibroid xenograft mouse model and examined its effects on tumor weight and gene expression. Bay is a phenyl vinyl sulfone that inhibits the NF-kB pathway by blocking the kinase activity of IKBKB and inhibiting the NLRP3 inflammasome [13,14]. It is also an inhibitor of protein tyrosine phosphatases and p65 phosphorylation [14]. Although not FDA-approved for any specific indication, Bay has been widely used in various in vivo and in vitro experimental disease models. This drug has broad anti-inflammatory effects in various disease models [14] and induces apoptosis in various cell types, including cancer cells [15,16].

## 2. Materials and Methods

### 2.1. Fibroid Specimens Collection

Portions of intramural uterine fibroids (2–5 cm in diameter) were collected from hysterectomies performed at Harbor-UCLA Medical Center for symptomatic fibroids (n = 10). Approval was obtained from the institutional review board (18CR-31752-01R) at the Lundquist Institute. Samples were exclusively sourced from premenopausal patients who had not received hormonal medications for at least 3 months prior to surgery. The fibroids used in this study were sourced from African American (n = 2) and Hispanic (n = 8) women aged 35–50 years (mean 43 ± 4.8 years). MED12 mutation analysis was conducted on all fibroids using PCR amplification and Sanger sequencing (Laragen Inc., Culver City, CA, USA). Among the tumors examined, five were found to harbor missense mutations in MED12 exon 2, specifically: c.130G>A (p.Gly44Ser) (n = 1), c.131G>A (p.Gly44Asp) (n = 2), c.130G>T (p.Gly44Cys) (n = 1), and c.131G>T (p.Gly44Val) (n = 1). Informed consent was obtained from all participants. A segment of each tumor was allocated for in vivo studies, while the remaining tissues were promptly snap-frozen and stored in liquid nitrogen for subsequent analysis, following established protocols to minimize variance among fibroids as previously described [17,18,19].

### 2.2. Fibroid Animal Model

The protocol (31162-02) received approval from the IACUC at the Lundquist Institute, Harbor-UCLA Medical Center. Female ovariectomized SCID/beige mice (Charles River Laboratories, Hollister, CA, USA), aged 9–12 weeks, were implanted with pellets (Innovative Research of America, Sarasota, FL, USA) containing estradiol (0.075 mg/60-day release) and progesterone (75 mg/60-day release), following established methods [20,21]. A 0.5 g portion of fresh fibroid tissue was aseptically sectioned into 5–10 pieces using a razor blade. Equal-weighted explants from the same patient were implanted into the flank of mice, which were then subsequently treated with Bay or vehicle, allowing for comparison of each treated tumor to its respective control. Following 3 days of recovery, mice received daily intraperitoneal injections of either vehicle (1% DMSO) or Bay (20 mg/kg). The dosage of Bay was determined based on previous studies demonstrating efficacy in an adult T-cell leukemia (ATL)/lymphoma mouse model [22]. After two months, all mice were anesthetized with ketamine (100 mg/kg) plus xylazine (10 mg/kg) via intraperitoneal injection, followed by cardiac puncture for blood collection and cervical dislocation for euthanasia. Xenografts were dissected free of surrounding tissue, weighed, and frozen. Animals were weighed before and after treatment with Bay or vehicle. Surgical procedures and experiments were conducted in the C.W. Steers Biological Resources Center (BRC) at the Lundquist Institute.

### 2.3. Blood Chemistry Panel

After 8 weeks of treatment with either the vehicle or Bay, plasma chemistry was analyzed. Specifically, 100 μL of plasma was transferred into the comprehensive diagnostic profile rotor (#500-1038, Abaxis, Union City, CA, USA). Subsequently, the levels of glucose, creatinine, phosphorus, BUN, sodium, alkaline phosphatase, albumin, globulin, serum glutamic pyruvic transaminase (SGPT; ALT), total protein, total bilirubin, and amylase were assessed using the VetScan VS2 chemistry analyzer (Abaxis, Union City, CA, USA) as previously described [23]. 

### 2.4. RNA Sequencing and Bioinformatic Analysis

Total RNA was extracted from leiomyoma and matched myometrium using TRIzol (Thermo Fisher Scientific Inc., Waltham, MA, USA). RNA concentration and integrity were evaluated using a Nanodrop 2000c spectrophotometer (Thermo Scientific, Wilmington, DE, USA) and an Agilent 2100 Bioanalyzer (Agilent Technologies, Santa Clara, CA, USA) as previously described. Samples with RNA integrity numbers (RIN) equal to or greater than 9 were selected for library preparation. One microgram of total RNA from each tissue was utilized to generate strand-specific cDNA libraries following the TruSeq protocol (Illumina, San Diego, CA, USA). The RNA sequencing and analysis were carried out at UCLA Technology Center for Genomics & Bioinformatics. Flaski facilitated the visualization of differential gene expression through volcano plots, hierarchical clustering, TreeView graphs, and pathway enrichment analysis [24]. Protein–protein interaction networks were constructed using the Search Tool for the Retrieval of Interacting Genes (STRING) database [25]. Overall, the obtained RNA sequencing data met the criteria for rigorous statistical analysis. Data have been deposited in the Gene Expression Omnibus (GEO) database under accession number GSE267860. 

### 2.5. Quantitative RT-PCR

Quantitative RT-PCR was performed as previously described [26,27]. Selected gene expression levels were assessed utilizing the Applied Biosystems 7500 Fast Real-Time PCR Systems, with normalization against 18S and RNU6B. Triplicate reactions were conducted, and relative expression was calculated using the comparative cycle threshold method (2^−ΔΔCt^), with results presented as fold change relative to the control group. The primer sequences used are shown in Appendix A. 

### 2.6. Immunoblotting

Proteins extracted from the xenografts underwent immunoblotting as previously described [28]. Primary antibodies targeting COL3A1, FN1, SPARC, TDO2, and cleaved caspase-3 were purchased from Proteintech Group, Inc. (Chicago, IL, USA). Band densities of specific proteins were quantified using the Image J program Version 1.51 (http://imagej.nih.gov/ij/, accessed on 26 March 2024) and normalized to a Ponceau S staining band on the membrane. Results were expressed as means ± SEM and are presented as ratios relative to the control group, designated as 1.

### 2.7. Enzyme-Linked Immunosorbent Assay

The total collagen content in xenografts was assessed in duplicate using the QuickZyme Total Collagen Assay Kit (QuickZyme Biosciences, Leiden, The Netherlands), following established protocols [20]. Absorbance readings were taken spectrophotometrically at a wavelength of 570 nm, and concentrations were determined by comparing sample optical density values to a standard curve. Total collagen levels were quantified as μg/mL of protein and reported as fold change relative to the vehicle group. 

### 2.8. Immunohistochemistry

Tissues were fixed in 4% paraformaldehyde and subsequently embedded in paraffin, and 5-micron sections were cut and mounted onto glass slides. Staining procedures, including Masson’s trichrome stain (HT15-1KT, Sigma-Aldrich, Burlington, MA, USA), and immunohistochemistry, were conducted as previously described [20,21,23]. Primary antibodies utilized in this study included rabbit anti-Ki67 (dilution 1:250, 27309-1-AP, Proteintech Group, Inc., Rosemont, IL, USA), rabbit anti-CCND1 (dilution 1:50, #55506, Cell Signaling Technology, Danvers, MA, USA), rabbit anti-cleaved caspase-3 (dilution 1:50, #9664, Cell Signaling Technology, Danvers, MA, USA), and mouse anti-E2F1 (dilution 1:50, sc-251, Santa Cruz Biotechnology, Inc., Dallas, TX, USA).

### 2.9. Fibroid Explant Culture 

Equal weights of fibroid explants, aseptically cut from the same patient, were plated in 6-well plates containing complete medium. They were then incubated for 48 h with either vehicle or Bay (5 μM).

### 2.10. Statistics 

The data was analyzed using GraphPad Prism 10 software (GraphPad, San Diego, CA, USA). Normality was assessed using the Kolmogorov-Smirnov test, and as the data were not normally distributed, nonparametric tests were employed for analysis. Wilcoxon matched pairs signed rank test was used for comparisons between two groups, while correlation analysis utilized the Spearman test. Statistical significance was established at *p* < 0.05. 

## 3. Results

Bay was well-tolerated with no effects on the body weight of mice and no changes in blood chemistry, including liver, kidney, and pancreatic function (Table 1). The administration of Bay led to 50% reduction in tumor weight after two months of treatment (Figure 1A,B).

The results of RNAseq on the xenografts from eight mice (four treated with vehicle and four with Bay) after two months of treatment are shown in Figure 2. Xenografts from the same patient were exposed to either vehicle control or Bay, thus allowing a matched comparison. As shown in the heat map (Figure 2A) and volcano plot (Figure 2B), Bay altered the expression of numerous genes, mostly inhibiting their expression. Using a cutoff of 1.5-fold, 374 transcripts were downregulated, and 629 transcripts were upregulated. The heat map (Figure 2C) shows 119 enriched hub transcripts identified using the CytoHubba plugin of the Cytoscape software platform version 3.10.2. Kyoto Encyclopedia of Genes and Genomes (KEGG) and Gene ontology (GO) enrichment analysis (Figure 2D) indicated that the differentially expressed genes primarily fell in the categories of cell proliferation and processes related to it, such as mitotic spindle organization and kinetochore, inflammatory and immune responses, and cytokine–cytokine receptor interaction. Network analysis indicated the complex potential interaction among the altered genes (Figure 2E). As shown in Figure 2C, the vast majority of genes were downregulated and were primarily involved with cell proliferation, mitosis, mitotic apparatus, and cytokinesis (e.g., *Ki67*, *E2F1*, *CCND1*, *CCNB1*, *CCNA2*, *BUB1*, *CKS2*, and others). The drug induced a decrease in the expression of a number of growth factors known to be upregulated in fibroids (e.g., *IGF1*, *EGF*, *PDGF*, *FGF8*, *PRL*, and *TGFB3*), and several components of the ECM, such as *COL3A1* and *KRT10*. The drug also induced downregulation of several proinflammatory cytokines known to be overexpressed in fibroids, including *IL1α and 1β*, *TNF*, *IL6*, *IL15*, *IL7R*, and *TNFRSF11A* (also known as RANK), and downregulation of *TLR3* and *TLR6* (whose function is the activation of NF-kB), and a number of chemokines (*CXCCR4*, *CCR5*, *CCL3*, and *CCL5*). In addition, Bay significantly inhibited the expression of *TDO2*, which we recently reported to be highly overexpressed in fibroids [20,29,30], and related to the tryptophan catabolic pathway and the serotonin receptor *HTR1B*. There was significant variation among the animals in the upregulated genes, with a lack of consistency among the xenografts. Notable genes were *vWF*, *ICAM1*, *ITGAX*, *SOX2*, *CXCL8*, *IL10*, and *IL13*.

Based on the RNAseq data, we proceeded to perform validation studies using qRT-PCR (Figure 3), Western blot analysis (Figure 4), and IHC (Figure 5) for select genes. As shown in Figure 3, Bay induced a significant decrease in the mRNA expression of genes involved in inflammatory response (*SPARC*, *TDO2*, *MYD88*, *TLR6*, *TLR3*, *IL6*, *TNFα*, *TNFRSF11A*, and *IL1β*, but not *IL8*), ECM remodeling (*COL3A1*, *FN1*, *LOX*, and *TGFB3*), cell cycle regulation (*CCND1*, *E2F1*, and *CKS2*), genes involved in hormone signaling (*PRL*, *ESR1*, and *PGR*), and cell signaling (*VEGFC*, *PDGFA*, and *HTR1B*). Due to limited sample availability, we could only perform Western blot analysis for a limited number of genes (shown in Figure 4A,B). As demonstrated in this Figure, Bay inhibited the protein abundance of COL3A1, FN1, SPARC, and TDO2 but not cleaved caspase-3. We also measured total collagen protein levels in the xenografts (Figure 4C). As shown, there was a significant decrease in total collagen protein levels in the xenografts of Bay-treated animals as compared with controls. Further confirmation studies were performed using IHC followed by Image analysis using the Halo software (Indica Labs-Area Quantification v2.4.2) in sections of xenografts (Figure 5). There was a significant reduction in the expression of the proliferation marker Ki67 in the drug-treated mice (Figure 5A,B), and decreased expression of several cell cycle markers including CCND1 (Figure 5C,D), and E2F1 (Figure 5E,F), which could account for the reduction in tumor weight in response to Bay. In line with Western blot analysis, we did not detect a significant difference in cleaved caspase-3 staining in drug- versus vehicle-treated controls (Figure 5G,H). Masson’s trichrome staining of xenograft sections revealed a significant decrease in the expression of collagens in Bay-treated xenografts and no significant differences in smooth muscle content in the two groups (Figure 5I,J). 

In vitro experiment was performed to determine which of the observed in vivo effects of Bay on gene expression are due to the direct effect of Bay on gene transcription. As shown in Figure 6, treatment of human fibroid explants with Bay (5 μM) after 48 h induced a significant inhibition of *VEGFC*, *TDO2*, *TGFB3*, *LOX*, *E2F1*, *PRL*, *PGR*, *CKS2*, *SPARC*, *COL3A1*, *FN1*, *CCND1*, *IL8*, *IL6* and *IL1β* mRNA levels without significant effects on *TLR3* and *TLR6*, *MyD88*, *TNFα*, *ESR1*, *HTR1B*, and *TNFRSF11A*.

Based on our prior studies indicating an interaction between NF-kB and miR-29c and miR-200c both of which are downregulated in fibroids [10,11], we quantified the expression of these miRNAs in the fibroid xenografts and human fibroid explants shown in Figure 7. As shown in this figure treatment with Bay led to a significant increase in the expression of miR-29c and miR-200c in the xenografts and fibroid explants. 

We performed a correlation analysis between fibroid weight and the expression of various genes shown in Figure 3 and Figure 6. As demonstrated in Figure 8 there was a significant positive correlation between tumor weight after two months of treatment with Bay or vehicle and the expression of *FN1*, a major component of the ECM [31], the inflammatory transcription factor *MyD88* [32], and miR-29c, a major regulator of ECM genes [10] but not any other genes shown in Figure 3 and Figure 6.

## 4. Discussion

The results of this study indicate that in vivo administration of Bay, the NF-kB inhibitor is well tolerated and highly effective in inhibiting the growth of fibroids, decreasing tumor weight by 50% after two months of treatment. This response to the drug was independent of the MED12 mutation status of the fibroid or the race/ethnicity of the patients. This decrease in tumor weight is primarily due to Bay’s effects on inhibiting the expression of genes regulating the cell cycle and cell proliferation, including *CCND1*, *E2F1*, *CKS2*, and *Ki67*. Bay also significantly reduced ECM accumulation, a hallmark of uterine fibroids by inhibiting the expression of collagens, including *COL3A1*, *FN1*, and *LOX*. The drug was also effective in inhibiting the expression of a number of genes involved in the immune response and inflammation (*SPARC*, *TDO2*, *MyD88*, *TLR3* and *6*, *IL6*, *TNFα*, *TNFRFSF11A*, *IL1β*) in the fibroid xenografts. Bay significantly decreased the expression of ESR1/PGR, which are critical for the growth and progression of fibroids [33,34], and a number of growth factors known to be upregulated in fibroids including prolactin [35,36], VEGF [37,38], PDGFA [39]. The effectiveness of Bay to shrink fibroid tumors by 50% and its induction of a favorable gene profile to inhibit genes associated with the cell cycle, ECM, and inflammation makes this and other drugs with similar modes of action a promising novel therapy for fibroids.

There is an established connection between inflammation and tumorigenesis. As such aberrant activation of NF-kB has been reported in multiple cancer [40]. Activation of NF-kB results in upregulation of genes regulating cell proliferation and migration and downregulation of genes regulating apoptosis [40,41,42]. This transcription factor also plays a key role in regulating the expression of genes in epithelial-mesenchymal transition [43] and cancer cell stemness [44]. Because of the role of NF-kB in oncogenesis, there has been significant interest in targeting it for cancer therapy [41,43,45]. Fibroid tumors are characterized by increased cell proliferation and increased expression of cell cycle proteins [46]. Our RNAseq data indicated that in vivo administration of Bay had the greatest effect on genes associated with cell cycle and cell proliferation in the fibroid xenografts. Confirmatory analysis showed decreased expression of a number of these genes, including CCND1 (cyclin D1) which regulates G1/S transition and is overexpressed in a variety of tumors [47]; E2F1, a transcription factor and cell cycle regulator that can mediate both cell proliferation and apoptosis [48], and CKS2, which binds to cyclin-dependent kinases and is essential for their function [49]. Bay has been used in experimental models of gastric cancer, where it induced S phase arrest by inhibiting Cyclin A and CDK-2 expression [50]. Similar inhibitory effects of this drug on cell proliferation were reported in multiple myeloma cells [15] and mantle cell lymphoma B cells [51].

Previous reports in various types of cancer have shown the induction of apoptosis by Bay [15,16,51,52]. However, in our in vivo study, Bay did not induce apoptosis in the fibroid xenografts as determined by cleaved caspase-3 protein levels and IHC analysis. There were significant variations among the xenografts, with some exhibiting increased apoptosis. This unexpected finding could be secondary to the drug dose used, which may not have been high enough to induce apoptosis, and the limited sample number of animals.

A hallmark of fibroids is excess accumulation of ECM [1]. Our data indicated that Bay reduced the expression of the main components of ECM, namely collagen and fibronectin. The reduction in tumor weight could be secondary to decreased ECM accumulation, as reflected by the positive correlation between tumor weight and FN1 levels. A critical regulator of fibrosis is the TGF-β cytokine family, which is produced by diverse cells and with critical roles in immunity [53], fibrosis [54], and cancer [55]. TGF-β3 is upregulated in fibroids and is critical in the overproduction of ECM in fibroids [56,57,58,59]. Bay significantly reduced the expression of TGF-β3 in the fibroid xenografts, which could be one of the mechanisms for the reduction in collagen and FN1 levels. Several studies have indicated a crosstalk between the TGF-β and NF-kB pathways [60,61,62]. TGF-β/SMAD was shown to be regulated by the interaction of SMAD3 with IKKα in breast cancer cells [63]. Similarly, TGF-β was shown to activate NF-kB by TGF-β activated kinase 1. This results in the phosphorylation of IKBα, leading to proteasomal degradation of IKBα and the release of NF-kB p65/p50 heterodimer and NF-kB activation [64]. Another gene we validated and related to ECM remodeling was Lysyl Oxidase (LOX), which was inhibited by Bay. LOX is a copper-dependent amine oxidase that catalyzes the cross-linking of collagen and elastin, thus contributing to ECM stiffness [65]. LOX not only plays a role in physiologic processes like organ development [66], but also in pathologic processes such as tumorigenesis [67,68] and progression of fibrosis [69]. The induction of NF-kB by advanced glycation end products was shown to upregulate lysyl oxidase expression in endothelial cells [70]. LOX also interacts with a number of growth factor signaling pathways identified in this study, including VEGF, PDGF, and TGF-β [71]. To our knowledge to date, there are no reports on the effect of Bay on LOX expression.

The growth of fibroids is dependent on the stimulatory effects of estrogen and progesterone [1]. We previously reported increased expression of total PGR protein and mRNA and PGRA protein in fibroid as compared with matched myometrium, and this was race-dependent, with greater expression in tumors from black patients [17]. Similarly, both ERα and ERβ are expressed in greater abundance in fibroids as compared with myometrium [33,72,73]. The inhibition of estrogen and progesterone receptor expression by Bay provides a significant beneficial effect of this drug for tumor growth, making them less responsive to the mitogenic effect of these ovarian steroids. Previous studies have shown a direct negative interaction between NF-kB and estrogen, progesterone, and glucocorticoid receptors [74]. Furthermore, both progesterone and glucocorticoid activated IKBα promoter, thereby inhibiting NF-kB in breast cancer cells [75], and estradiol-induced estrogen response element activity in endometrial cells by an NF-kB dependent and estrogen receptor mechanism [76]. The in vivo effect of Bay on estrogen/progesterone receptor expression has not been reported to date. Bay also inhibited the expression of PRL in the xenografts. Several studies have indicated that PRL is expressed and secreted by fibroid and myometrial tissue and that it stimulates cell proliferation [35,36]. A more recent study indicated that PRL may have a role in the differentiation of fibroblasts into myofibroblasts, which actively contribute to ECM accumulation [35]. Thus, the reduction in PRL expression by Bay is another beneficial effect of this drug to reduce fibroid growth and ECM volume. Fibroids are also characterized by increased angiogenesis with increased microvessel density and VEGF expression [77]. Bay effectively inhibited VEGFC mRNA expression in fibroid xenografts. Other studies have shown the importance of NF-kB in tumor angiogenesis and the role of VEGF in this process [77]. PDGF is another growth factor that promotes cell proliferation and was previously reported to be upregulated in fibroids [78]. In vivo administration of Bay significantly reduced the expression of PDGF. Previous studies showed the role of NF-kB in regulating PDGF expression in fibroblasts [79]. Little is known about serotonin and its receptors in fibroids. In our previous study examining the tryptophan catabolic pathway, we did not find any differences in serotonin levels in fibroids versus myometrium [30]. However, in another study, a selective blocker of 5HT1B receptor decreased fibroid cell proliferation, induction of apoptosis, and reduced cyclin D1 and αSMA expression [80]. Collectively, the inhibition of expression of growth factors such as PRL, PDGF, HTR1B, and VEGF in fibroid xenografts provides another mechanism for reducing tumor cell proliferation and angiogenesis in fibroids.

The inhibition of canonical NF-kB by Bay expectedly inhibited the expression of proinflammatory cytokines IL6, IL1β, and TNFα. These cytokines are also activators of canonical NF-kB [81], and among them, IL1β and TNFα are overexpressed in fibroids [6,82]. Bay also inhibited several genes upstream of the NF-kB pathway, including TLRs and MyD88. Toll-like receptors are pattern recognition receptors that through adaptor proteins such as MyD88, convey messages to the NF-kB complex. TLR6 is a membrane-associated receptor whereas TLR3 is endosome-associated [83,84]. Activation of TLR6 receptors results in its interaction with MyD88, and activation of TLR3 with the protein TRIF, both of which lead to the activation of NF-kB [85]. There is only one report indicating activation of NF-kB and TLR4 in fibroid cells and uterine fibroid-derived fibroblast [86], and no published reports on MyD88 in fibroids. Thus, Bay not only inhibits NF-kB but also upstream inputs that activate the canonical NF-kB, including proinflammatory cytokines TNFα and IL1β, toll-like receptors, and their adaptor proteins through yet unidentified mechanisms.

In vivo treatment with Bay also inhibited SPARC, a matricellular protein that plays a key role in tissue repair and wound healing and has a wide range of functions in ECM production, inflammation, cell cycle, growth factor activity, and cell adhesion [87]. SPARC is overexpressed in inflammatory conditions such as rheumatoid arthritis and different types of cancer [87]. Our previous report showed a marked overexpression of SPARC mRNA, which was dependent on the MED12 mutation status of fibroid [26]. In bone, NF-kB activation was shown to stimulate SPARC [88], and SPARC inhibited the metabolic programming of ovarian cancer cells and adipocytes by an NF-kB-mediated mechanism [89]. Our group first showed a marked dysregulation of tryptophan catabolism in fibroids, with a marked overexpression of TDO2, which also was race and MED12 dependent [29,30]. Its inhibition in vitro [29] and in vivo [20] resulted in decreased fibroid cell proliferation, decreased ECM accumulation, and reduced inflammation in fibroids. Several studies have provided evidence of crosstalk between TDO2 and the NF-kB pathway [90,91,92]. Our current data supports the existence of this crosstalk, showing inhibition of NF-kB by Bay reduces the expression of TDO2, which would contribute to the inhibition of fibroid cell proliferation, ECM production, and inflammation.

In this study, we measured the expression of two key miRNAs in fibroid pathogenesis, both of which are downregulated in fibroids namely miR-29c, which regulates many components of the ECM such as collagens, and miR-200c, which regulates many of the cell cycle and inflammation regulatory proteins [1]. Our previous research revealed that treatment with Bay and silencing p65 by siRNA both induced the expression of miR-29c and miR-200c in primary fibroid smooth muscle cells [10,93], indicating the significant involvement of NF-kB in their regulation. We have also presented evidence indicating that treatment with tranilast reduces p65 nuclear translocation, thereby diminishing its binding capacity to the miR-200c promoter, leading to the induction of miR-200c [93]. Other studies have shown that miR-29c is epigenetically silenced by an activated NF-κB/YY1 complex, which functions as a suppressor of the miR-29c promoter in rhabdomyosarcoma [94]. In this study, we show that treatment of fibroid xenografts and human fibroid explants with Bay significantly increases the expression of both miR-29c and miR-200c. The overexpression of these miRNAs would be expected to result in decreased expression of their target genes, including those associated with cell cycle regulation, ECM, and inflammation. The lack of change in IL8, a target of miR-200c in this study, was unexpected but could be due to secondary effects of Bay or inadequate dose of drug. 

In vitro experiments with fibroid explants, which preserve all the cell types, were used to determine which of the in vivo effects of Bay on gene expression are due to direct effects of NF-kB on gene transcription. With the exceptions of TLR3, TLR6, MyD88, and TNFRSF11A, all of the observed inhibitory effects of Bay in the in vivo study were shown to be at least in part due to a direct involvement of NF-kB during their transcription. Epigenetic mechanisms, such as the altered expression of some miRNAs (as shown for miR-29c and miR-200c) or growth factors, could be other mechanisms for the indirect effects of NF-kB inhibitor on gene transcription. 

Various drugs that target the complex NF-kB pathway at different levels of the pathway are in development for the treatment of inflammatory conditions, such as rheumatoid arthritis, autoimmune diseases, and cancer, and some of these drugs have already gained FDA approval [7,85]. Our study indicates the usefulness of inhibiting the canonical NF-kB pathway using Bay for the treatment of fibroids. In a previous in vivo study, we showed that tranilast, which is approved for the treatment of asthma and fibrotic diseases in Asia, also has beneficial effects for the treatment of fibroids by reducing tumor size, inhibiting collagen expression and cell proliferation, and increasing cellular apoptosis [23]. In vitro studies indicated that tranilast works by inducing the expression of miR-200c, which in turn inhibited the NF-KB signaling pathway by preventing nuclear translocation of RelA-p65 [93]. The noncanonical NF-kB could also be targeted for fibroid treatment, as shown in a previous study wherein its inhibition by RANK-Fc in vivo blocked RANKL-induced expression of cyclin D1, decreasing fibroid growth and expression of Ki67 [95]. Our study indicates that inhibition of the canonical NF-kB pathway via Bay also indirectly inhibits TNFRSF11A, which is upregulated in fibroid stem cells [95].

## 5. Conclusions

In summary, this preclinical study indicates that in vivo administration of Bay is well tolerated and without side effects. The inhibition of the canonical NF-kB pathway has a beneficial effect in reducing the growth and progression of human fibroids in a mouse xenograft model. This reduction in tumor weight is accompanied by the induction of a highly favorable gene expression profile in the fibroid xenografts, with reduced expression of cell cycle regulatory proteins, components of the ECM such as collagen and FN1, proinflammatory cytokines (IL6, IL1β, and TNFα), estrogen and progesterone receptor, growth, and angiogenic factors (PDGF, VEGFC, SPARC, and TDO2) and induction of two miRNAs with critical roles in fibroid pathogenesis (namely, miR-29c and miR-200c). Bay also inhibited genes upstream of NF-kB signaling, including TLR3, TLR6, and MyD88, as well as the noncanonical activator TNFRFSF11A (RANK). In vitro experiments with Bay in fibroid explants indicated that the in vivo inhibitory effects of Bay on these genes were probably due to direct inhibitory effects of NF-kB, with the exceptions of TLR3 and TLR6, MyD88, and TNFRSF11A. These promising data warrant further exploration of the NF-kB pathway for fibroid treatment, and potentially repurposing of some already approved drugs. 

## Figures and Tables

**Figure 1 cells-13-01091-f001:**
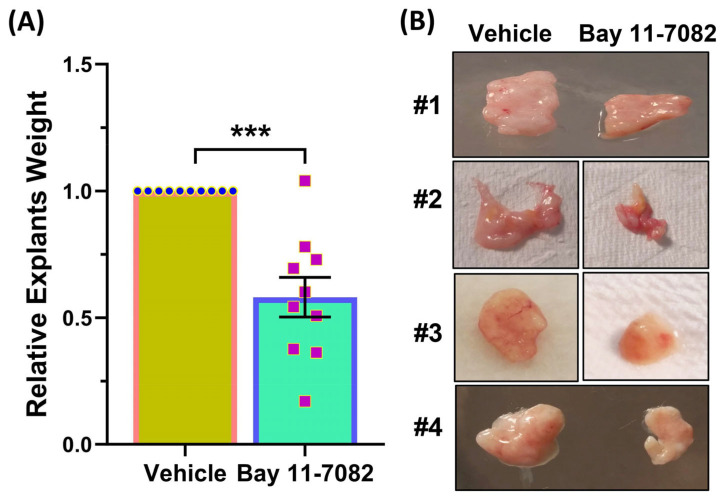
(**A**) Fresh fibroids explants were subcutaneously implanted in ovariectomized CB-17 SCID/Beige mice, followed by daily intraperitoneal administration of vehicle or Bay 11-7082 (20 mg/kg) for 8 weeks. Tumor explant weights were measured at the end of the 8-week treatment period (n = 10). (**B**) Representative images of four xenografts at the end of treatment period (8 weeks). Data are presented as mean ± SEM of independent experiments, with corresponding *p* values indicated on the respective line. *** *p* < 0.001.

**Figure 2 cells-13-01091-f002:**
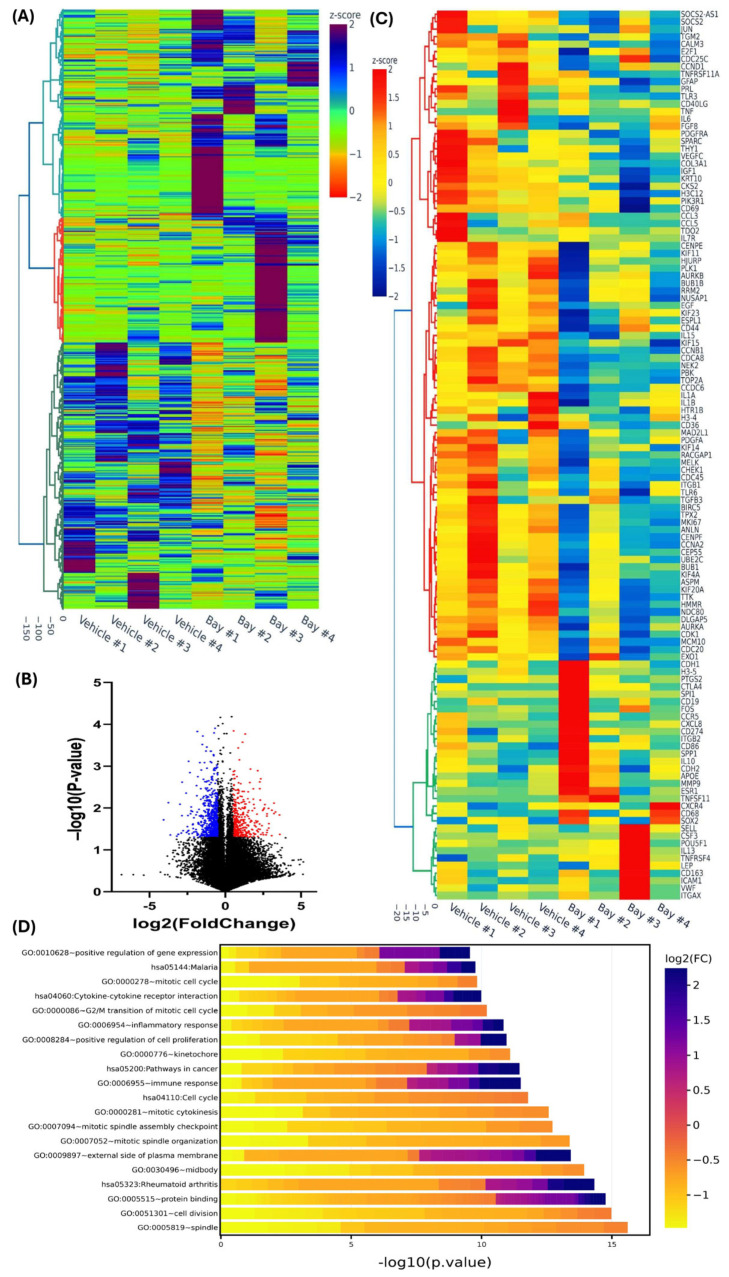
(**A**) Hierarchical clustered heatmap analysis depicting differentially expressed transcripts (fold change ≥ 1.5, *p* < 0.05) in xenografts after 8 weeks of treatment with vehicle or Bay 11-7082 (n = 4). Color gradient represents gene expression as z-scores. (**B**) Volcano plot illustrating upregulated (n = 374; red) and downregulated genes (n = 629; blue) with a false discovery rate (FDR) *p*-value < 0.05. (**C**) Heatmap displaying the expression profiles of 118 enriched hub genes identified by the CytoHubba plugin in Cytoscape. Gene expression levels are represented as z-scores. (**D**) Gene ontology (GO) analysis of the 118 enriched hub genes, presented as log2-fold change levels across a color gradient. (**E**) Protein-Protein Interaction Networks of the 118 hub genes constructed using the Search Tool for the Retrieval of Interacting Genes (STRING) database and Cytoscape software version 3.10.2. Node colors indicate interaction degree (red, orange, yellow, from high to low degree).

**Figure 3 cells-13-01091-f003:**
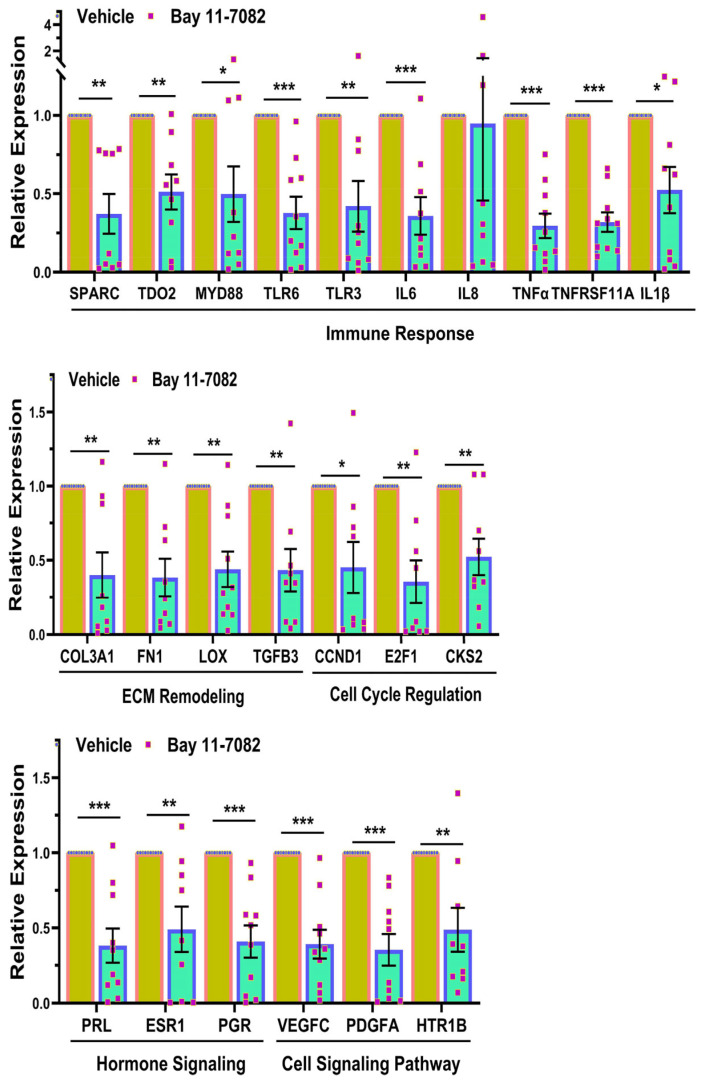
Relative mRNA expression levels of SPARC, TDO2, MYD88, TLR6, TLR3, IL6, IL8, TNFα, TNFRSF11A, IL1β, COL3A1, FN1, LOX, TGFB3, CCND1, E2F1, CKS2, PRL, ESR1, PGR, VEGFC, PDGFA and HTR1B in subcutaneously implanted xenografts in ovariectomized CB-17 SCID/Beige mice (n = 10) following 8 weeks of treatment with vehicle or Bay 11-7082 (20 mg/kg/daily). Data are presented as mean ± SEM, with *p* values indicated on respective lines. * *p* < 0.05; ** *p* < 0.01; *** *p* < 0.001.

**Figure 4 cells-13-01091-f004:**
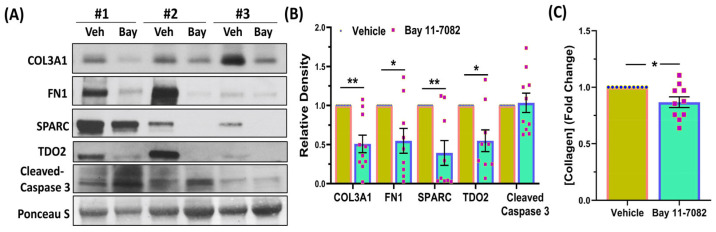
(**A**) Representative Western blot analysis of COL3A1, FN1, SPARC, TDO2, and cleaved caspase 3, with corresponding bar graphs (**B**) showing their relative band densities in the xenografts (n = 10). (**C**) Total collagen levels assessed by enzyme-linked immunosorbent assay in the 10 xenografts. Data are presented as mean ± SEM of independent experiments, with *p* values indicated on the respective line. * *p* < 0.05; ** *p* < 0.01.

**Figure 5 cells-13-01091-f005:**
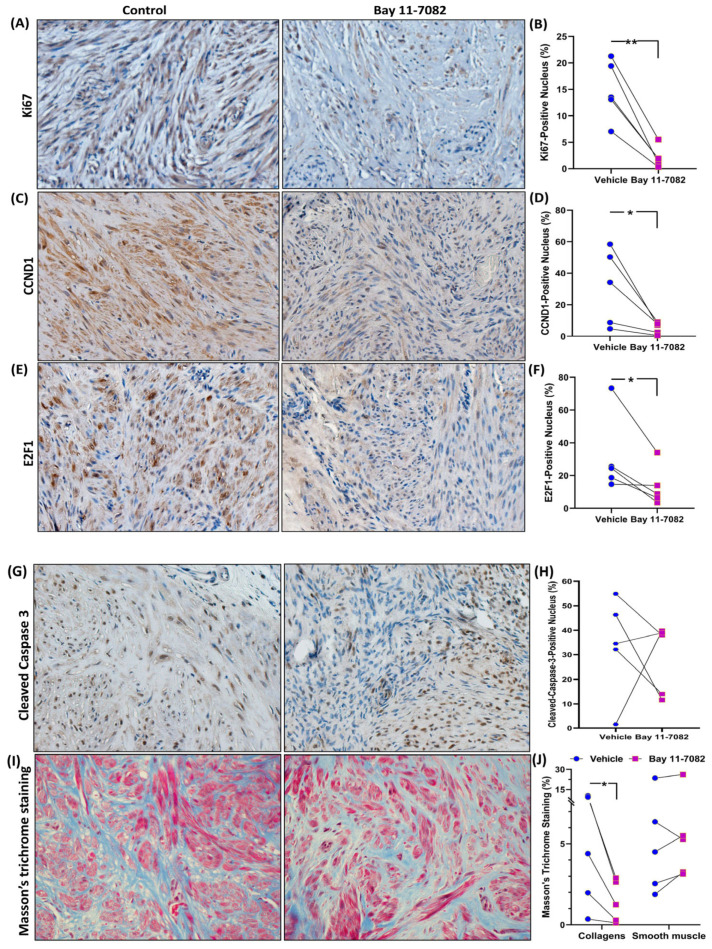
(**A**–**H**) Representative immunohistochemically stained images of fibroid xenografts treated with vehicle or Bay 11-7082 (magnification, ×20) for Ki67 (**A**,**B**), CCND1 (**C**,**D**), E2F1 (**E**,**F**) and cleaved caspase-3 (**G**,**H**). Quantification of positive staining in the nucleus by Halo software (Indica Labs-Area Quantification v2.4.2) is shown in panels ((**B**,**D**,**F**,**H**); n = 5 in each group). (**I**) Representative histopathological images stained with Masson’s trichrome of fibroid xenografts from the vehicle or Bay 11-7082-treated group (magnification, ×20). Blue color indicates collagen fibers, while red color represents smooth muscle cells. (**J**) Quantification of staining intensity by Halo software (n = 5 in each group). Data are presented as mean ± SEM, with *p* values indicated on the respective line. * *p* < 0.05; ** *p* < 0.01.

**Figure 6 cells-13-01091-f006:**
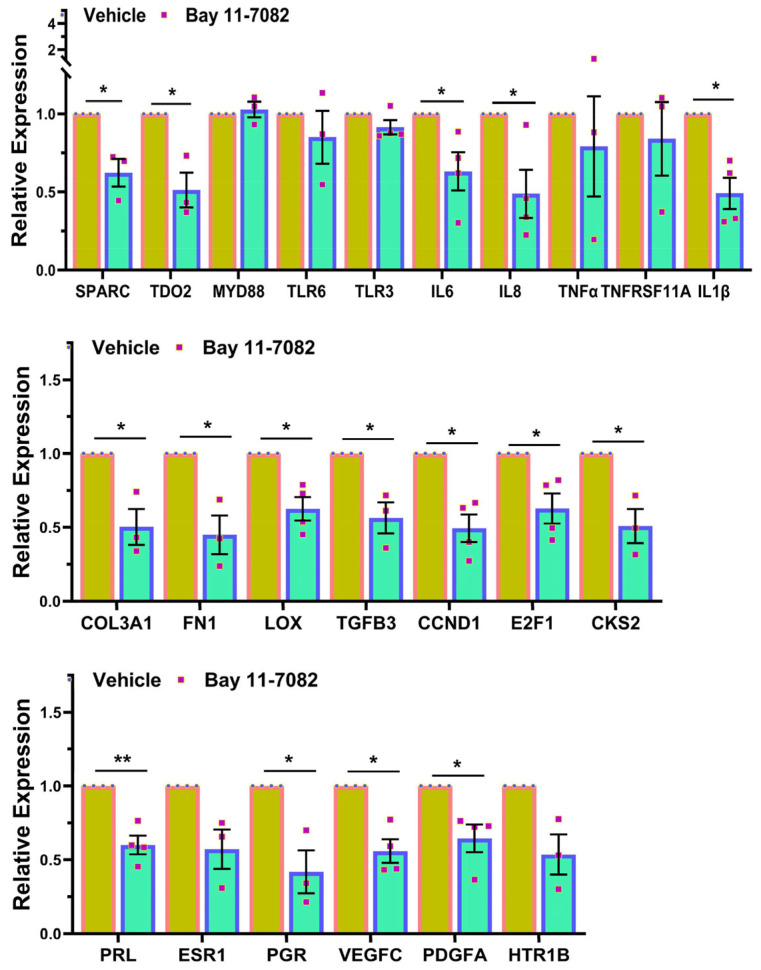
Relative expression of SPARC, TDO2, MYD88, TLR6, TLR3, IL6, IL8, TNFα, TNFRSF11A, IL1β, COL3A1, FN1, LOX, TGFB3, CCND1, E2F1, CKS2, PRL, ESR1, PGR, VEGFC, PDGFA and HTR1B mRNA in fibroid explants (n = 4) following 48 h of treatment with vehicle or Bay 11-7082 (5 μM). The results are presented as mean ± SEM with *p* values indicated by corresponding lines. * *p* < 0.05; ** *p* < 0.01.

**Figure 7 cells-13-01091-f007:**
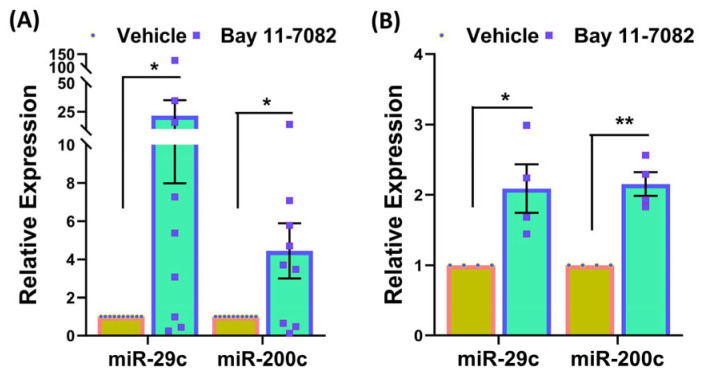
Relative expression of miR-29c and miR-200c in subcutaneously implanted xenografts in ovariectomized CB-17 SCID/Beige mice (n = 10) following 8 weeks of treatment with vehicle or Bay 11-7082 (20 mg/kg/daily) (**A**), and in fibroid explants (n = 4) following 48 h of treatment with vehicle or Bay 11-7082 (5 μM) (**B**). Data are presented as mean ± SEM, with *p* values indicated on the respective lines. * *p* < 0.05, ** *p* < 0.01.

**Figure 8 cells-13-01091-f008:**
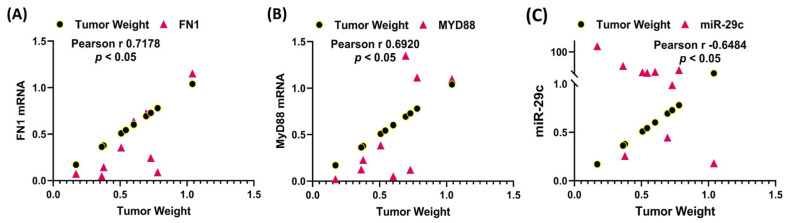
Correlation analysis between the fold change in tumor weight with levels of FN1 (**A**), MyD88 (**B**), and miR-29c (**C**) in the xenografts.

**Table 1 cells-13-01091-t001:** Chemistry panel of mice after two months of treatment with Bay 11-7082.

Chemistry Panel Marker	Vehicle	Bay 11-7082	*p*-Value
	(Mean ± SEM)	(Mean ± SEM)	
General metabolism			
Glucose (mg/dL)	230.9 ± 17.73	261.0 ± 14.63	0.069
Kidney function			
BUN (mg/dL)	24.3 ± 2.8	27.1 ± 5.75	0.648
Creatinine (mg/dL)	0.2 ± 0.29	0.4 ± 0.03	0.939
Electrolytes			
Sodium (mEq/L)	154.4 ± 0.98	155.3 ± 1.35	0.628
Phosphorus (mg/dL)	8.7 ± 0.77	10.1 ± 0.53	0.988
Liver function			
Alkaline phosphatase (U/L)	105 ± 16.95	91.9 ± 20.05	0.355
Albumin (g/dL)	3.9 ± 0.12	3.5 ± 0.14	0.063
SGPT (ALT) (U/L)	44 ± 10.3	54.5 ± 8.71	0.504
Total protein (g/dL)	5.2 ± 0.61	4.6 ± 0.16	0.326
Globulin (g/dL)	1.3 ± 0.5	1.1 ± 0.24	0.786
Total bilirubin (mg/dL)	0.2 ± 0.02	0.3 ± 0.04	0.732
Pancreas function			
Amylase (U/L)	703.9 ± 32.93	643.0 ± 36.82	0.074

## Data Availability

Raw data were generated at The Lundquist Institute. Derived data supporting the findings of this study are available from the corresponding author O.K. on request.

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
