# Peer review of "In Vivo Effects of Bay 11-7082 on Fibroid Growth and Gene Expression: A Preclinical Study"

_cells, 2024, doi:10.3390/cells13131091_

Round 1

Reviewer 1 Report

Comments and Suggestions for Authors

Chuang et al investigated the potential application of Bay 11-7082 (Bay) in treating fibroid using a xenograft mouse model. They showed that Bay led to reduced tumor weight and altered expression of genes related to cell proliferation, ECM remodeling, and inflammation. Overall, the study is well structured, and the authors performed detailed experiments to validate results from their RNA-seq experiments. The study is of value to the community, and I have only a few minor comments summarized below:

1.    As the authors mentioned in their introduction, Bay is a broad-spectrum inhibitor. Therefore, calling Bay 11-7082 an NF-kB inhibitor in the title (and elsewhere) is inaccurate and can be misleading. The authors need to perform extra experiments (for example, Bay treatment in NF-kB KD/KO cells) if they wish to claim a causal relationship between the in vitro/in vivo effect of Bay and NF-kB inhibition,

2.    The authors claimed to have performed ELISA (abstract, line 7) but I could not find such data.

3.    In the method section, the authors mentioned that the primer sequences are included in Supplementary Table 1. However, it seems that the table contains data from RNA-seq.

4.    All mRNA names should be italicized to distinguish between mRNA transcripts and proteins.

5.    Figure 2A/C can be improved: to make the heat map more informative, the authors might want to perform clustering analysis and group genes according to their behavior in gene expression changes between Veh and Bay-treated groups. Additionally, the authors should perform a GO analysis of the main clusters. The authors might also wish to rearrange the panels as it’s very hard to see panel B (the volcano plot).

6.    In addition to point#5, I wonder if the authors checked the overlap of the up/down-regulated genes with published datasets of NF-kB targets and transcripts known to be regulated by Bay?

7.    In Figure 4, have the authors also blotted for pro-caspase 3 level? To report on no change in Caspase 3 activity, the authors should also show that the total amount of Caspase 3 remains unchanged.

8.    A description of the expression analysis of miRNAs is lacking in the method section.

9.    Figure 8 is confusing. Does each dot represent values from either Tumor weight or levels of each mRNA, or both? It would also be great if the authors could plot the line of fit.

Author Response

Reviewer #1
Chuang et al investigated the potential application of Bay 11-7082 (Bay) in treating fibroid using a xenograft mouse model. They showed that Bay led to reduced tumor weight and altered expression of genes related to cell proliferation, ECM remodeling, and inflammation. Overall, the study is well structured, and the authors performed detailed experiments to validate results from their RNA-seq experiments. The study is of value to the community, and I have only a few minor comments summarized below:

  1. As the authors mentioned in their introduction, Bay is a broad-spectrum inhibitor. Therefore, calling Bay 11-7082 an NF-kB inhibitor in the title (and elsewhere) is inaccurate and can be misleading. The authors need to perform extra experiments (for example, Bay treatment in NF-kB KD/KO cells) if they wish to claim a causal relationship between the in vitro/in vivo effect of Bay and NF-kB inhibition,

Response: Thank you for the useful suggestions. We have revised our title to “In Vivo Effects of Bay 11-7082 on Fibroid Growth and Gene Expression: A Preclinical Study” In our future studies, we will consider using NF-kB KD/KO cells to clarify the mechanisms of Bay 11-7082 in fibroids, both in vivo and in vitro.

  1. 2. The authors claimed to have performed ELISA (abstract, line 7) but I could not find such data.

Response: The data is shown in Fig. 4C.

  1. In the method section, the authors mentioned that the primer sequences are included in Supplementary Table 1. However, it seems that the table contains data from RNA-seq.

Response: The primer sequences are indeed included in Supplementary Table 1.

  1. All mRNA names should be italicized to distinguish between mRNA transcripts and proteins.

Response: Thanks for the suggestions. We have updated the format for mRNA transcripts.

  1. Figure 2A/C can be improved: to make the heat map more informative, the authors might want to perform clustering analysis and group genes according to their behavior in gene expression changes between Veh and Bay-treated groups. Additionally, the authors should perform a GO analysis of the main clusters. The authors might also wish to rearrange the panels as it’s very hard to see panel B (the volcano plot).

Response: Thank you for your suggestions. We have rearranged the panels to improve the resolution of Fig. 2. In fact, we created the heatmap based on the differential gene expression changes between Veh and Bay-treated groups (Fig. 2A). We then focused on the 118 enriched hub genes shown in Fig. 2C for their GO and protein-protein interaction networks analysis (Fig. 2D and E).

  1. In addition to point#5, I wonder if the authors checked the overlap of the up/down-regulated genes with published datasets of NF-kB targets and transcripts known to be regulated by Bay?

Response: Thank you for your suggestions. Unfortunately, we could not find any published RNAseq datasets of NF-kB targets regulated by Bay in fibroids at this time.     

  1. In Figure 4, have the authors also blotted for pro-caspase 3 level? To report on no change in Caspase 3 activity, the authors should also show that the total amount of Caspase 3 remains unchanged.

Response: Thank you for the suggestions. Since pro-caspase 3 is inactive in viable cells and is only activated after cleavage into cleaved-caspase 3, the levels of cleaved-caspase 3 reflect the apoptosis status. Unfortunately, we do not have any xenograft samples left in some sets. We will incorporate the suggested assay in our future experiments.

  1. A description of the expression analysis of miRNAs is lacking in the method section.

Response: Thanks for the suggestions. We have updated it in the method section as below:

“Quantitative RT-PCR was performed as previously described [26, 27]. Selected gene expression levels were assessed utilizing the Applied Biosystems 7500 Fast Real-Time PCR Systems, with normalization against 18S and RNU6B, respectively. Triplicate reactions were conducted, and relative expression was calculated using the comparative cycle threshold method (2-ΔΔCt), with results presented as fold change relative to the control group. The primer sequences used are shown in Supplementary Table 1.”

  1. Figure 8 is confusing. Does each dot represent values from either Tumor weight or levels of each mRNA, or both? It would also be great if the authors could plot the line of fit.

Response: Yes, each dot represents values from either tumor weight or mRNA levels as shown in different colors.

Reviewer 2 Report

Comments and Suggestions for Authors

Comments to the authors

Thank you for inviting me to review your wonderful work! I acknowledge the hard work behind this preclinical study, and I truly believe that your manuscript should be accepted after minor changes.

Here are my comments and indications:

Comment 1: The introduction section is comprehensive, but I suggest splitting the large paragraph into smaller ones because it is hard to read. For example, “In earlier work we reported activation of NF-kB in fibroids…” should constitute a single paragraph, the test hypothesis another paragraph, and the aim of the study the last paragraph. Also, there are some extra spaces that need to be removed.

Comment 2: The materials and methods section is very well detailed and structured. The small number of subjects is not unusual for this type of study. I have only one observation: have you managed to obtain the accession number for the GEO database? If yes, please insert it in the main manuscript.

Comment 3: Table 1 from the results section needs to be converted into standard MDPI format, and should be accompanied by a legend. Also, the glucose levels of these mice are high. Could you comment on this observation? When I worked with Streptozotocin‐induced diabetic models in mice, their baseline glucose levels, before injection, were much lower.

Comment 4: In the results section, please be careful and describe all abbreviations when first used in text.

Comment 5: I know how difficult it is sometimes include a heatmap with all the details, but could you please increase the resolution of fig 2 A,B? Their details are hard to read.

Comment 6: The discussion section is complex, and the same problem with large paragraphs needs to be addressed. Other than that, some minor spelling errors are present, and I kindly invite the authors to carefully read the text.

Comment 7: The conclusion section is valuable, and I agree with their line of thinking.

Comments on the Quality of English Language

There are some minor spelling errors.

Author Response

Reviewer #2
Thank you for inviting me to review your wonderful work! I acknowledge the hard work behind this preclinical study, and I truly believe that your manuscript should be accepted after minor changes.

Here are my comments and indications:

Comment 1: The introduction section is comprehensive, but I suggest splitting the large paragraph into smaller ones because it is hard to read. For example, “In earlier work we reported activation of NF-kB in fibroids…” should constitute a single paragraph, the test hypothesis another paragraph, and the aim of the study the last paragraph. Also, there are some extra spaces that need to be removed.

Response: Thanks for the suggestions. We have edited the introduction section.

Comment 2: The materials and methods section is very well detailed and structured. The small number of subjects is not unusual for this type of study. I have only one observation: have you managed to obtain the accession number for the GEO database? If yes, please insert it in the main manuscript.

Response: Thanks for your support. We have updated the GEO accession number (GSE267860) in the method section.

Comment 3: Table 1 from the results section needs to be converted into standard MDPI format, and should be accompanied by a legend. Also, the glucose levels of these mice are high. Could you comment on this observation? When I worked with Streptozotocininduced diabetic models in mice, their baseline glucose levels, before injection, were much lower.

Response: Thank you for your suggestions. We have updated the format of Table 1. We also observed elevated glucose levels in both groups of mice, which may be attributed to the side effects of the implanted steroid pellets. Research indicates that an imbalance of estrogen and progesterone can affect cellular responses to insulin, potentially causing insulin resistance and resulting in blood sugar fluctuations (PMID: 33005004; 21768169). We have sacrificed all the mice and used their plasma for a blood chemistry panel assay. In future experiments, we will double-check this phenomenon.

Comment 4: In the results section, please be careful and describe all abbreviations when first used in text.

Response: Thanks for the suggestions. We have carefully checked and updated the full names of abbreviations when first used in the whole article.

Comment 5: I know how difficult it is sometimes include a heatmap with all the details, but could you please increase the resolution of fig 2 A,B? Their details are hard to read.

Response: Thanks for the suggestions. We have expanded figures 2A and B.

Comment 6: The discussion section is complex, and the same problem with large paragraphs needs to be addressed. Other than that, some minor spelling errors are present, and I kindly invite the authors to carefully read the text.

Response: Thanks for the suggestions. We have edited the discussion section.

Comment 7: The conclusion section is valuable, and I agree with their line of thinking.

Response: Thanks for your support.

Reviewer 3 Report

Comments and Suggestions for Authors

The manuscript is nicely written and well-organized. My main critique would be that the figures are too small to appreciate the fine details.  I suggest since space is not a limit for this journal expanding all figures to encompass the full length of the margins allowed in the template.

Table 1 should be increased by 30-50%.

Author Response

Reviewer #3

The manuscript is nicely written and well-organized. My main critique would be that the figures are too small to appreciate the fine details. I suggest since space is not a limit for this journal expanding all figures to encompass the full length of the margins allowed in the template.

Response: Thanks for the suggestions. We have expanded all figures.

Table 1 should be increased by 30-50%.

Response: Thanks for the suggestions. We have expanded the Table 1.